# A qualitative process evaluation of group problem management plus for distressed Syrian refugees in Türkiye

## Research Article

refugees; mental health problems; task-sharing; scale-up; process evaluation

**Corresponding author:**
Ceren Acarturk;
Email: cacarturk@ku.edu.tr

Ayşenur Coşkun-Toker[1] ⓘ, Zeynep İlkkurşun[1] ⓘ, Daniela Fuhr[2,3,4] ⓘ,
Bayard Roberts[4] ⓘ, Pim Cuijpers[5] ⓘ, Marit Sijbrandij[5] ⓘ and Ceren Acarturk[1]

[1]Department of Psychology, Koc University, Istanbul, Türkiye; [2]Research Group Implementation Research and Mental Health, Leibniz Institute of Prevention Research and Epidemiology, Bremen, Germany; [3]Health Sciences, University of Bremen, Bremen, Germany; [4]Department of Health Services Research and Policy, London School of Hygiene and Tropical Medicine, London, UK and [5]Department of Clinical, Neuro and Developmental Psychology, Amsterdam Public Health Research Institute, WHO Collaborating Center for Research and Dissemination of Psychological Interventions, Vrije Universiteit Amsterdam, Amsterdam, the Netherlands

## Abstract

Syrian refugees in Türkiye show a high prevalence of mental health problems but encounter barriers to accessing mental health services. Group Problem Management Plus (gPM+), developed by the World Health Organization, is a low-intensity psychological intervention delivered by nonspecialist facilitators. This qualitative process evaluation explores the acceptability, feasibility and perceived effectiveness of gPM+ for Syrian refugees resettled in Türkiye, as well as facilitating factors and barriers to its implementation. Twenty-three semi-structured interviews were conducted with gPM+ participants, facilitators, drop-outs, relatives of participants and key informants. Findings showed that gPM+ was well-received for its group-based format, which participants felt fostered social support, and for its content, which they reported may have led to improvements in coping skills and family relationships. Facilitators viewed the intervention as feasible to implement. However, barriers such as participants' economic struggles, practical challenges (e.g., childcare and transportation difficulties) and low mental health literacy impeded engagement. Adapting gPM+ to address social determinants like poverty may be beneficial. The need for booster sessions was emphasized to maintain long-term change and provide deeper learning of the strategies. For sustainable scaling up gPM+ within primary health care, key informants highlighted the importance of training and supervising nonprofessional facilitators and securing governmental support.

## Impact statement

Providing mental health care to refugees is a challenge to mental health systems and a pressing issue following the sudden increase in their numbers, as they are one of the most vulnerable yet underserved populations. Therefore, it is crucial to understand the barriers and facilitators of implementing mental health interventions for these populations to improve their access to mental health services. This qualitative process evaluation explores the underlying challenges and facilitating factors that influence implementation and outcomes of WHO's scalable intervention gPM+ for Syrian refugees in Türkiye. Through an in-depth understanding of these factors, this study aims to shed light on what works – and what does not – in a real-world setting for accessing and benefiting from mental health interventions among refugees. Drawing on the perspectives of different stakeholders, it provides practical recommendations for mental health professionals, decision-makers and relevant authorities in order to improve the implementation and scale-up of mental health care for refugees in low-resource settings. This study also deepens our understanding of gPM+ and its adaptability to the needs of refugee populations. Ultimately, this study provides valuable insights to enhance the accessibility and effectiveness of mental health care for refugee populations worldwide.





## Introduction

Over 110 million individuals globally were forcibly displaced because of conflict, violence or human rights abuses in mid-2023 (UNHCR, 2023). As a result of the Syrian war, over 6.6 million people fled to neighboring countries. Of those, Türkiye hosts 3.6 million Syrian refugees (UNHCR, 2022). Refugees have frequently experienced violent and traumatic events, and ongoing daily stressors including discrimination, loss of social support and financial problems. All these contribute to the risk of developing common mental health problems including

depression, anxiety and post-traumatic stress disorder (PTSD) (Morina et al., 2018). Research shows a high prevalence of mental health problems among Syrian refugees in Türkiye. In refugee camps in Türkiye, the prevalence of symptoms of PTSD and depression was 83.4% and 37.4% respectively (Acarturk et al., 2018). In a setting in Istanbul, the prevalence of PTSD, depression and anxiety symptoms among Syrian refugees was 19.6%, 34.7% and 34.7% respectively (Acartürk et al., 2020).

Despite the burden of common mental health problems, Syrian refugees in Türkiye have restricted access to and underutilization of mental health services, with only approximately 8% of those affected receiving treatment (Fuhr et al., 2020a). This can be attributed to system-level factors such as the cost of mental health services, the lack of sufficient mental health professionals and interpreters, the absence of cultural adaptation in treatments and societal factors such as mental health stigma (Fuhr et al., 2020a; Kazdin, 2021). Moreover, in the standard model of mental health care delivery (including in Türkiye), treatments are delivered in person in a one-to-one format at private clinics by a mental health professional. This highly specialized approach is challenging to implement in low-resource settings (Tol et al., 2012), due to scarcity of professionals (WHO, 2022), a lack of flexibility in adopting diverse treatment approaches for comorbid conditions (Bryant, 2015) and financial barriers that limit access for individuals without sufficient economic resources (Bolton et al., 2007).

Efforts to fill this care gap and to improve access to evidence-based interventions are crucial for improving mental health care, especially in lower-resource settings. To support this, the World Health Organization (WHO) has developed simplified, brief versions of evidence-based mental health interventions. These interventions seek to be scalable and are characterized by being low-intensity, transdiagnostic and utilizing a task-shifting approach that allows trained and supervised nonprofessionals to deliver them (WHO, 2017). One prominent intervention among these is Problem Management Plus (PM+), which aims to treat common mental health problems based on evidence-based behavioral principles. It consists of four evidence-based strategies: (1) stress management, (2) problem solving, (3) behavioral activation and (4) increasing social support (Dawson et al., 2015). PM+ is a 5-session intervention that can be implemented by trained and supervised nonspecialist facilitators in individual and group formats. A meta-analysis showed that both individual and group PM+ (gPM+) had a significant effect on reducing mental health problems and improving positive mental health outcomes, both of which remained significant in long-term assessments (Schäfer et al., 2023). Within the STRENGTHS project, which aimed to scale up psychological interventions for Syrian refugees, gPM+ has been found effective in reducing depression symptoms and personally identified problems among Syrian refugees in a camp in Jordan, but showed no significant effect on symptoms of PTSD, anxiety and functional impairment (Bryant et al., 2022). As part of STRENGTHS, gPM+ was evaluated in Türkiye among Syrian refugees and led to a reduction in depression and anxiety for participants with higher baseline distress in the short-term, and significantly improved functional impairment at the 3-month follow-up. However, for the overall sample, it did not significantly reduce symptoms of anxiety, depression and PTSD (Acartürk et al., 2024).

Exploring the impact and the scalability of a mental health intervention and guiding the process of large-scale implementation requires an assessment not only of the effectiveness of the intervention but also of the acceptability and feasibility of intervention delivery. Understanding the experiences and perceived impact of those receiving the service, as well as potential barriers and facilitators

of intervention delivery, may be crucial to inform efforts to further maintain and to scale up an intervention (Massazza et al., 2022). This paper presents findings from qualitative interviews conducted with stakeholders involved in a definitive randomized controlled trial (RCT) to evaluate the effectiveness of gPM+ among 368 adult Syrian refugees in Türkiye (Acartürk et al., 2024). The present study aimed to explore the acceptability, feasibility and perceived impact of gPM+ and identify potential barriers and facilitators to implementing gPM + as perceived by different stakeholders.

This qualitative process evaluation was informed by the Medical Research Council's (MRC) guidance for process evaluation of complex interventions (Moore et al., 2015), which outlines three key components that guided our interview questions and analysis. Accordingly, we examined (1) how gPM+ was delivered (implementation), (2) how stakeholders experienced and responded to gPM+ (mechanisms) and (3) how contextual factors influenced its delivery and uptake (context).

## Methods

### Study setting and participants

This study was a qualitative process evaluation conducted in the final phase of a definitive RCT evaluating the effectiveness of the gPM+ among Syrian refugees in Türkiye (see Supplementary Material S1 for details), as part of the STRENGTHS project (Sijbrandij et al., 2017). Semi-structured interviews were conducted with 23 respondents. These were (a) three female and two male gPM+ participants, (b) three male and two female gPM+ participants who dropped out, (c) four male and one female relatives of gPM+ participants who completed or dropped out of intervention, (d) two female and two male gPM+ facilitators who delivered the intervention and (e) four key informants, including a project coordinator, program manager, psychologist and social worker. gPM+ participants were older than 18 years, had temporary protection status, were Arabic speaking and showed elevated levels of psychological stress (measured by the General Health Questionnaire-12) and impaired functioning (measured by the WHO Disability Assessment Schedule). gPM+ facilitators were nonprofessional Syrian or Egyptian Arabic-speaking peers with similar backgrounds, as they were also forcibly displaced people. They had completed a minimum of 12 years of education, and most were university students, with no prior experience in delivering psychological support. Key informants were the professionals who have a relevant role, such as program implementation, supervision and coordination, within the Refugees and Asylum Seekers Assistance and Solidarity Association (RASASA) – a non-governmental organization serving Syrian refugees, with whom the trial was implemented in collaboration.

The sample size of 23 was determined based on purposive sampling – aimed at capturing diverse perspectives across stakeholder groups (e.g., participants, facilitators, relatives and key informants). Participant type (i.e., completers or drop-outs) was also used as a purposive criterion to receive different insights regarding the acceptability and feasibility. Within these groups, we aimed for variation in terms of gender and age. Both male and female facilitators were included to reflect the gender-matched delivery structure of gPM+. Key informants (i.e., RASASA professionals) were selected based on their experience in delivering and managing mental health and psychosocial support (MHPSS) services for Syrian refugees and their familiarity with the gPM+. This enabled them to provide informed observations on key aspects of service delivery within this refugee community and system-level challenges. While a purposive sampling strategy guided the initial

selection of the sample, we also monitored the thematic saturation – the point at which no new themes were emerging. This was ascertained when summarizing interview data. None of the selected participants declined to participate in the interviews. Data were collected by a team of independent interviewers who were not involved in the RCT and had no prior relationship with any of the participants. These were university graduates or final-year undergraduate students who were fluent in Arabic and English.

Semi-structured interviews followed a topic guide involving several open-ended questions, including acceptability and feasibility of gPM+, barriers and facilitating factors for treatment engagement, views on the group format and the facilitators, views of gPM+ facilitators on barriers and facilitating factors for delivering gPM+, barriers and facilitating factors for large-scale implementation and views on the benefits and challenges of integrating gPM+ into primary health care services (see Supplementary Material S2). The topic guide was tailored to the specific group being interviewed. Interviews were conducted in Arabic by native Arabic-speaking researchers who were also fluent in English. They later translated the field notes into English themselves to minimize the risk of meaning loss during translation. Due to COVID-19 restrictions, interviews could not be conducted face-to-face. Interviews with gPM+ participants, drop-outs and their relatives were conducted by telephone to accommodate their limited internet access, while interviews with facilitators and key informants were conducted via video conferencing. In all interviews, it was confirmed that participants were in a private space. Each interview lasted approximately 45–60 min and was not audio recorded due to the governmental regulations of the Immigration Authority of Turkey, which does not permit making audio recordings in research studies with Syrian refugees. So, the interviewers were responsible for taking field notes during the interviews. They were trained in the use of qualitative interviewing methods, including how to take detailed field notes.

### Data analysis

We conducted our analysis based on the Framework Method (Gale et al., 2013), a systematic and flexible approach well-suited to applied health research. We followed seven structured steps. First, the verbatim notes from interviews conducted in Arabic were translated into English by bilingual research assistants and transcribed. Second, after the familiarization with the data, we recorded initial reflective notes on their content. Third, we coded the interviews following both deductive and inductive approaches. Deductive codes were initially derived from the semi-structured interview guide (i.e., topic guide), which reflects predefined areas of interest. Inductive coding was also used to identify new emerging themes that were not initially considered. Fourth, after coding a subset of transcripts independently, broader themes and related subcodes were identified, which formed the basis of an analytical coding framework. Each theme included multiple codes and subcodes, with quotations linked to each subcode (see Supplementary Material S3). Fifth, the agreed codes were applied to all transcripts using NVivo 11 (QSR International, 2015). Thirteen out of twenty-three interviews were double-coded by two researchers. A total of 121 coding decisions were compared, with 99 matches and 22 mismatches, resulting in an agreement percentage of 81.82% and Cohen's Kappa of 0.89, which indicates almost perfect agreement. The inconsistencies were discussed and resolved. Sixth, we charted the data into a framework matrix, with each row representing a quote categorized under a specific subcode, code and theme, alongside participant identifiers.

Lastly, we interpreted the matrix to identify patterns across different respondents.

This manuscript was reported in accordance with the COREQ 32-item checklist for qualitative studies (Tong et al., 2007) (see Supplementary Material S4).

### Results

In this section, the term 'participants' refers exclusively to individuals involved in the gPM+ trial, while 'facilitators' refers to those who delivered the gPM+ intervention. For other respondents, their identities or titles, such as 'family member' or 'project coordinator,' are used.

### *Acceptability and perceived effectiveness of gPM+*

The participants (i.e., gPM+ trial participants) stated that their experiences with gPM+ were generally positive, with most expressing satisfaction and benefits from the program. They reported feeling that the strategies they learned were helpful in identifying and coping with personal problems, providing support in stressful times and improving family relationships. One participant stated that *"It was a good program and I enjoyed it as my soul my tired and this helped me identify the problem and resolve it."* Additionally, their relatives observed improvements in participants' psychological well-being and their relationships after the intervention. One of the participants' husbands stated, *"This affected the family in a positive way as she was trying to mingle in the new society and this made her mood get better as well as the mood of the whole family."* While some participants pointed out the positive impact of their family's support and encouragement in their attendance to the program, others encountered challenges due to familial concerns, such as childcare responsibilities. The participants' experiences and acceptability of the gPM+ were also impacted by their perceived need for the intervention. Some participants who dropped out expressed skepticism about the program's necessity for themselves as the reason for discontinuing.

Many participants and their relatives stated that the participants appeared to apply the skills they learned in their lives. They reported perceiving positive outcomes of gPM+ strategies, such as becoming more socialized and engaged in activities they previously avoided. However, not all participants noticed significant differences, and some expressed challenges in consistently applying the skills. One participant mentioned that she needed more sessions to practice these skills with the facilitator. One family member noted that the intervention's positive impact did not persist after the completion of the sessions, even if there was progress during the intervention. Some of the participants who dropped out were skeptical about the effectiveness of the program. An issue mentioned was financial problems, as the husband of a participant who dropped out stated, *"I will be honest with you, as long as we have financial problems, nothing will be beneficial to us."*

The acceptability and perceived effectiveness of the gPM+ were also acknowledged by the gPM+ facilitators. They noted that participants often provided positive reports regarding how they applied techniques in real life and how the program positively impacted their lives. One facilitator noted that approximately 70% of participants appeared to apply the strategies in their lives, especially problem-solving and behavioral activation. She considered the problem-solving strategy to be the most impactful, especially for those who were emotionally overwhelmed post-war

and who struggled to solve problems logically. Another facilitator reported that the breathing technique appeared to be frequently applied and was believed to be beneficial by some participants as a pause from life's stresses. Social strategies were also observed to be effective for participants in enhancing social interactions, and were reported as well-accepted by participants, since they align with the Syrian culture.

### Feasibility of attending gPM+ sessions

Participants reported experiencing challenges in attending the gPM+ sessions, mostly because of daily responsibilities, particularly childcare. Female participants generally faced difficulties in leaving their children to come to the sessions, and their spouses did not support their participation due to the need to look after the children. On the other hand, despite initial concerns about childcare, some family members noticed benefits, as one husband stated, "*I thought it would be a problem at first because she should take care of our child but actually her attendance made a positive impact on the family.*" Regarding the feasibility of attending sessions, some drop-outs reported transportation challenges, such as the absence of direct public transportation opportunities. Many completers reported no difficulties in attending the sessions and found the timing suitable. Only one participant noted a lack of follow-up communication from the program organizers, resulting in missed sessions.

### Group format and gPM+ facilitators

Participants generally expressed positive experiences with the group format of the intervention. They noted that other participants and facilitators were good; they made new friends and had a good time. The social interaction within the group was perceived as a beneficial experience that contributed to participants' well-being and facilitated good bonding. Participants enjoyed the chance to interact with others and this improved their mood. One participant's husband stated, "*She was happily going to the program and she was speaking with the group, which made her mood get better, so yes, there was a good bond as I see.*" Only one participant, who dropped out, mentioned that he faced challenges with the group. He explained, "*The people who attended the program were always asking inappropriate questions that have nothing to do with the program.*" Additionally, another participant suggested that one-to-one sessions might be much more beneficial to participants than group sessions, even if the latter were not bad.

Facilitators suggested that the group format may have strengthened a sense of community and empowerment among participants. One facilitator shared the following feedback from a participant: "*I learned something to solve my problem, but there is another power with me—not only the facilitator but also the group.*" One facilitator attributed the positive impact on participants' well-being not only to the strategies they learned but also to the supportive environment provided by the group. The motivational aspect of group dynamics, where motivated participants inspired others to engage in sessions and try the strategies, was also emphasized. The project coordinator also expressed that the group setting of gPM+ was a facilitating factor for refugees in Türkiye, who usually live collectively and may not seek individual help due to cultural and religious reasons.

Participants' views regarding the facilitators of gPM+ were all positive. They appreciated that the facilitators managed the group well, and did not report any problem in communication in terms of culture and language. All participants acknowledged the competence of the facilitators, and depicted them as supportive, helpful

and kind. One participant expressed her view about the facilitator, saying, "*This was the first time I feel that someone cares about us and our suffering.*" All drop-outs also noted positive views about facilitators, despite not completing the program.

### Facilitating factors of gPM+ delivery

Facilitators reported several factors that helped participants stay motivated to complete the program. First of all, the facilitators highlighted the importance of building rapport with participants. They emphasized their role in participants' lives as someone who listens, tries to help, provides useful techniques and follows up with them. This helped to foster a positive and trusting relationship, encouraging participants to continue. He also noted that regular interaction and meetings support a trustful relationship and facilitate participants to feel more engaged, by stating, "*Most of them were very isolated in their houses and so scared of the world outside. Then they started going out on a regular basis for these weekly meetings. We build some trust between them and us.*" Facilitators also emphasized the importance of the first session to build rapport and to create a strong first impression by effectively explaining the program.

Tailoring the delivery of the intervention to fit the different educational and cognitive levels of the participants was identified as another facilitating factor. Facilitators reported that they adapted their communication and materials to the needs of different demographics. The facilitators also shared their personal motivations and positive experiences, underscoring the satisfaction derived from making a meaningful impact on participants' lives. They described their experience as gratifying and rewarding, despite emotional challenges when witnessing participants' suffering.

### Barriers to gPM+ delivery

Facilitators identified various barriers that impacted the delivery and the perceived effectiveness of the gPM+. Some of these were related to participants' lack of genuine interest and engagement. One facilitator noted, "*I remember one of them said, 'I think what you are doing here is useless, it would be better if you gave the money instead.'*" Facilitators also encountered difficulties when participants' expectations from the program did not align with the gPM+ content. For instance, some participants insisted on focusing on emotional problems that are unsolvable, and facilitators found it challenging to bring the conversation back to relevant topics without ignoring the participants' desire.

Another major barrier was related to the participants' mental health literacy. Facilitators expressed their concerns about whether participants have the intellectual capacity to fully grasp and apply the strategies they learned in the program, considering their educational backgrounds. One facilitator noted, "*The background of the participants matters in terms of how much they will get from the implementation or how fast they will get it. These were people who did not know how to read and write. We had to repeat ourselves, refine, draw and give examples.*" Another facilitator noted that the knowledge and awareness of mental health was low within the cultural context of the group, and shared, "*I would be very reductionist if I say it is merely about education. The whole mental health thing is not really heard of in the community. You are giving details about something that people do not have the basics of.*" He mentioned that some participants were skeptical about strategies like breathing exercises, which led to personal discomfort or resistance, while he suggested that a social support enhancement strategy may

be more familiar within the Syrian culture and may lead to more acceptance. Facilitators also pointed out the difficulties of participants in adopting new habits in five sessions, especially for the elderly ones.

The structure of gPM+ also posed some barriers. Most facilitators reported that the five-week duration of the intervention might be insufficient, especially for participants with trauma backgrounds. Facilitators mentioned that it took a longer time – until almost the third session – for participants to build trust and open up; therefore, more sessions were needed to cover the content on a deeper level. One facilitator said, "*We were receiving complaints about the program being short. Because it took too long for them to give their all, actually focus and think, and to feel safe. Because you know they are coming from trauma so, to feel safe and feel that this is a safe place to talk…*" One facilitator also emphasized the need for more practice and support for the application of strategies in real-life and long-term behavior change. Another facilitator noted the need for regular reminders or follow-up sessions for sustaining long-term improvement. Facilitators also pointed out the need for referrals to individual sessions for discussing unsolvable problems that are beyond the scope of this intervention and for developing supplementary programs.

Some barriers related to adherence of the intervention were also highlighted, with some facilitators expressed concerns about applying the strategies long-term and in real-life situations. Despite participation and engagement in sessions, facilitators pointed out that some participants did not practice the strategies in their daily lives. Negative comments or resistance from their family members also weakened participants' commitment or caused them to drop out. One facilitator noted, "*They learn something in the session, but someone who doesn't know the intervention made negative comments like 'this will not work'… They were saying, 'I told my husband, and he said that it's a piece of cake, you are having fun.' I think this can demotivate them too.*"

Lastly, facilitators reported some barriers related to contextual factors affecting the Syrian refugee community in Türkiye. Syrian refugees in Sultanbeyli, a low-income area, endure harsh conditions and face a lack of future prospects. Facilitators expressed that these adverse life conditions often lead to feelings of hopelessness among participants, who stated that without changes in their life conditions, psychological interventions seemed useless. One facilitator noted the sentiments of a participant, stating, "*He was like, 'you guys are giving us this intervention, but nothing is changing in conditions. Change our condition and we would feel better instead of trying to make us feel better in the same condition.*'" The anger and hopelessness expressed by refugees, stemming from their living situation, also compounded the emotional burden of facilitators and led to questioning the efficacy of their efforts.

### Scaling up gPM+

All of the key informants agreed that gPM+ can be effectively delivered in primary health care (PHC) settings. The project coordinator noted, "*gPM+ can be delivered very well in primary health care services such as municipal psychological support centers and mother-child centers.*" They emphasized that it is crucial to inform decision-makers, such as relevant staff in the Ministry of Health, about the benefits and feasibility of gPM+ to garner support for integration into PHCs. The project coordinator suggested presenting the outcomes of the project and concrete data to convince decision-makers. The psychologist recommended starting with pilot studies in certain PHCs to see the benefits of gPM+ and create high demand for

broader acceptance. The social worker pointed out the importance of a good organization of the staff who will deliver gPM+ and adequate supervision for an effective integration.

Key informants also identified potential barriers for scaling up gPM+ in PHC settings. The project coordinator defined the general structure of the public sector as cumbersome and noted that changing the perceptions of older decision-makers and acceptance of new initiatives may be challenging. The lack of multilingual staff, particularly for projects involving refugees, can be another barrier. Limited access to PHC services for individuals in poorer areas or with limited transportation options, as well as engaging men, especially those from lower socioeconomic backgrounds who work long hours, were considered additional potential barriers. The program manager mentioned that integration of gPM+ into PHC services can be challenging due to instructional resistance, such as reluctance among these services to adopt new interventions, and the requirement for extensive training of staff. Similarly, the psychologist noted that determining who will provide gPM+ can be a barrier due to a lack of acceptance among existing busy staff. Lastly, some barriers among refugees were identified, such as cultural perceptions and daily responsibilities., These included the caregiving roles of women, long work hours for men, stigma related to mental health services and a lack of support from family members for their participation.

### Integration of gPM+ into primary health care

Key informants emphasized skepticism of decision-makers regarding the credibility and effectiveness of nonprofessionals in delivering a psychological intervention as a barrier that may hinder the acceptance and integration of gPM+. The psychologist recommended integrating gPM+ training into psychology graduate programs and assigning graduates to PHC settings. The social worker noted that the integration of external personnel as facilitators into the PHC system can be challenging, so training existing staff would be more feasible. The psychologist suggested that gPM+ can be an initial step in a wider referral system. Moreover, reframing PM+ as a course to help with personal problems rather than as a psychological intervention was recommended to reduce prejudice and encourage participation. The program manager also recommended reframing gPM+ to enhance acceptance of nonspecialist facilitators. Key informants also suggested that gPM+ can be integrated not only into PHC services but also into workplaces or municipalities, which include psychological support centers.

### Discussion

The findings of this study showed that gPM+ was generally well-received by participants and feasible to implement, aligning with previous studies showing that PM+ is acceptable, feasible and safe among Syrian refugees (de Graaff et al., 2020; Akhtar et al., 2021; Acarturk et al., 2022; Spaaij et al., 2022). However, several intervention-specific and context-specific challenges were identified. While gPM+ participants, their relatives and facilitators commonly reported perceived improvements in participants, those who dropped out were skeptical about the intervention's effectiveness, due to the greater urgency of addressing their economic problems. This finding underscores the potential adverse effect of financial struggles on an individual's ability to engage with psychological interventions, similar to the results from the previous qualitative evaluations of PM+ conducted in low-resource settings (van't Hof et al., 2018; Ali et al., 2023).

Researchers have highlighted the need to address post-displacement stressors, such as poverty and unemployment, to support the engagement and effectiveness of psychological interventions among refugees (Miller, 1999; Goodkind et al., 2013). One example is a multisectoral integrated interventions that combine mental health support with financial and employment assistance (Miller and Rasmussen, 2017; Evans et al., 2022).

Practical challenges such as childcare responsibilities or transportation difficulties were identified as attendance barriers, especially for women, reflecting findings from elsewhere (Byrow et al., 2020; Woodward et al., 2023). To reduce opportunity costs, some adjustments could be considered in the future implementations of gPM+, such as partnering with local community organizations to offer shared childcare services or providing more accessible locations. Furthermore, supportive or discouraging attitudes of families were identified as factors that can either facilitate or obstruct a participant's attendance, reflecting findings from previous studies (van't Hof et al., 2018). Therefore, informing families about the potential benefits of gPM+ before the program could increase family support, which, in turn, may improve participant attendance.

Facilitators' competence, attitude, ability to adapt content to participants' background and cultural sensitivity were reported as key factors for the successful delivery of gPM+ and participants' engagement. This accords with evidence highlighting the importance of facilitators' personal qualities and trust-building abilities in implementing psychosocial programs (Dickson and Bangpan, 2018), and their potential to effectively create caring environments and implement scalable interventions (van't Hof et al., 2018; Le et al., 2022). Several barriers were identified, including some participants' lack of genuine interest and differing expectations about the intervention. Before the program, providing a more comprehensive orientation that clarifies the intervention's content and goals may help ensure participants have realistic expectations. Participants' limited mental health literacy also seemed to impede their ability to fully grasp and apply some strategies. A previous study conducted with Syrian refugees in Istanbul also found that limited mental health knowledge contributed to low mental health service utilization (Fuhr et al., 2020b). The findings highlight the need to raise awareness of mental health among Syrian refugees. Respondents also emphasized the need for additional sessions to sustain long-term change or achieve a deeper understanding. One way to address this could be offering booster sessions for those experiencing ongoing stressors or struggling to apply skills.

All the key informants emphasized the potential for scaling up gPM+ within PHC settings. The need for adequate training and supervision of nonspecialist facilitators was highlighted, consistent with previous studies (van't Hof et al., 2018; Spaaij et al., 2023). Future gPM+ implementations could benefit from standardized quality control systems, such as WHO's ensuring quality in psychological support (EQUIP) tool (Spaaij et al., 2023), in assessing competencies for effective psychosocial support and psychological care. Ultimately, our findings suggest that a sustainable scale-up of gPM+ requires the support of decision-makers and its integration into a centrally funded system, which regulates recruitment protocols, referral systems and ensures sustainable funding to support adequate training, supervision and remuneration of facilitators, as highlighted in previous studies (van't Hof et al., 2018; Spaaij et al., 2023; Woodward et al., 2023). Some studies indicated that these interventions could be sustainably scaled with long-term financial resources and cooperation between local NGOs and the government (Fuhr et al., 2020a). Particularly in LMICs, the co-financing approach – where mental health services are funded through partnerships between governments, NGOs or private investors – provides a sustainable model that strengthens national health systems (Woodward, 2023). The Turkish Ministry of Health could adopt such a model. Future research should investigate strategies to gain government support for the inclusion of gPM+ in local planning, how its implementation can be funded locally, how facilitators can be sustainably trained and supervised and how gPM+ can be framed and adapted for specific groups to enhance acceptance.

This study has some limitations. First, interviews were not audio-recorded due to regulatory restrictions, which may have limited the depth and completeness of the data despite the presence of detailed field notes. Second, interviews were conducted in Arabic, and translation into English may have resulted in the loss of some nuances in the language. Third, even though the interviews were conducted by independent researchers, there may have been some bias, as respondents could have given answers they believed the interviewer wanted to hear. Fourth, only Syrian refugees who agreed to participate in a psychological intervention and were living in Sultanbeyli District of Istanbul were interviewed. This may restrict the transferability of the results to other refugee contexts, such as camp settings. Lastly, member checking was not conducted, which may limit the confirmability of findings, since respondents did not have the chance to review or validate the researchers' interpretations of their input.

In conclusion, this study provides insights into the factors influencing the current delivery and potential scale-up of gPM+. However, these findings should be interpreted within the study's methodological and contextual limitations. Further research is needed across diverse settings and participants (e.g., rural or camp-based environments, non-NGO-led contexts such as public health care systems), and using additional qualitative validation strategies such as member-checking and audio-recording if possible. Such studies will inform much-needed implementation of scalable interventions in humanitarian contexts.

**Open peer review.** To view the open peer review materials for this article, please visit http://doi.org/10.1017/gmh.2025.10035.

**Supplementary material.** The supplementary material for this article can be found at http://doi.org/10.1017/gmh.2025.10035.

**Data availability statement.** The data of this study can be made available upon reasonable request to the corresponding author.

**Author contribution.** A.C.T., Z.I., D.C.F., B.R., C.A. contributed to conception of the study. Z.I. contributed to the data collection and A.C.T. and Z.I conducted the analysis of data. A.C.T., Z.I., D.C.F and C.A. conducted the interpretation of the data analysis. A.C.T. drafted the manuscript. D.C.F, B.R., P.C., M.S. and C.A. revised the manuscript critically, provided inputs for the interpretation of the findings and gave final approval of the version for submission.

**Financial support.** This work was supported by the European Union's Horizon 2020 Research and Innovation Programme Societal Challenges [grant number 733337]. The content of this article reflects only the authors' views, and the European Community is not liable for any use that may be made of the information contained therein.

**Competing interests.** The authors declare no conflict of interest.

**Ethics statement.** The study was approved by the Ethics Committees of Koc University (Protocol ID: 2021.025.IRB3.006), and the Immigration Authority of the Republic of Türkiye.

The research was conducted in accordance with the principles of the Declaration of Helsinki. All participants gave their written informed consent prior to participation.

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
