## [Reviewer Report]

A Qualitative Process Evaluation of Group Problem Management Plus for Distressed Syrian Refugees in Türkiye

This manuscript presents a qualitative evaluation on the delivery of Group Problem management Plus (gPM+) to Syrian refugees in Turkiye. This is an important study that seeks to understand the perspectives of key stakeholders who experienced gPM+ within different capacities of a definitive RCT that evaluated the effectiveness of gPM+ in Turkiye. To sustainably embed scalable programs like PM+ within local health systems, systematically understanding the lived experiences of such stakeholders is an important part of the bigger implementation piece. The following feedback relate to the methodology of this qualitative study and is intended to enhance the value of this work.

1) Recruitment and data collection: Can you please justify the final sample size of 23 stakeholders in terms of your approach to reaching this decision. I note that purposive sampling was used to select heterogenous participants relating to age and gender factors. How about other key characteristics? Specifically, did you engage in purposive sampling, or data saturation, or a combination of both, as it pertains to the content of the feedback received? It would be helpful for the readers to know how such decisions were made at each stage of the data synthesis (e.g., reading or summarising of interview data, generation of codes and construction of themes).

2) Key informant characteristics: It would help the reader to contextualise key informant perspectives within the context of recency of experience in delivering psychological programs, or MHPSS services, or being involved in the management and implementation of similar services. For example, when evaluating key informant perspectives on specific challenges regarding uptake of programs within PHCs, it is particularly important to purposefully sample participants who have experienced similar challenges. Please provide brief details on this. Additionally, did any of the key informants who were contacted decline to participate in the interviews. If so, could you provide details on reason for non-response.

3) Results: You ask some important questions regarding the scalability and integration of brief psychological programs within the local health system. Even so, there are significant reservations regarding the reliability and validity of the outputs given the current methods. Please elaborate further what attempts were made to ensure reliability and validity of data; for example, did you use data saturation principles to achieve this, or purposive sampling. If you did not make similar assurances during the course of the study, please acknowledge this and the related limitations in the conclusions that can be drawn and applicability to other settings.

4) Conclusions and limitations: I caution statements regarding the comprehensiveness of the factors investigated in the current study as it related to future scalability of gPM+ in Turkiye (lines 19-24). Please consider further methodological and applicability limitations regarding the context of your study, and how future research with refugees in Turkiye could overcome these limitations to increase the reliability and validity of the outputs reported here. The limitations currently reported in the manuscript do not sufficiently capture the concerns raised above.

---

## [Reviewer Report]

Aim of the study: This qualitative process evaluation explores the acceptability, feasibility, and perceived effectiveness of gPM+ for Syrian refugees resettled in Türkiye, as well as facilitating factors and barriers to its implementation. Twenty-three semi-structured interviews were conducted with gPM+ participants, facilitators, drop-outs, relatives of participants, and key informants.

General comments.

The study is an interesting addition to the much-missing information on the process evaluation of a gPM+, a widely utilized intervention in humanitarian settings. Such studies will help inform adaptation and implementation in related settings through learning offered from such qualitative studies.

However, the study will benefit from methodology revision to ensure complete reporting of the approach utilized in the study. Guidelines for the reporting of qualitative studies (COREQ) https://www.equator-network.org/reporting-guidelines/coreq/ or SRQR (https://journals.lww.com/academicmedicine/fulltext/2014/09000/Standards_for_Reporting_Qualitative_Research__A.21.aspx) will be helpful.

Specific comments

Before methodology, it would be appropriate to detail the process evaluation theoretical framework used. Check out the MRC guidelines for process evaluation (https://www.bmj.com/content/350/bmj.h1258) or other available alternatives. While they do not aim to ensure that all aspects of the evaluation framework are covered, they provide guidelines to help ensure comparability of process evaluation outcomes.

Line 56, page 5, what informed sample size of 23 participants? Its consideration depends on the type of qualitative inquiry you decide to take; hence, it will be good to highlight the qualitative study approach used.

Page 6, lines 49-52. it will be good to understand how notes were taken as interviews were not recorded. Also highlight any shortcomings due to this approach in the limitations section.

Page 7, line 3. Framework analysis means you used an existing framework or pre-determined codebook in your coding process. However, your analysis sections do not expound on this, making it difficult to understand what was done in line 26. As I highlighted earlier, provide a framework matrix to use for this. The process of framework analysis used is not explicitly clear from your methodology section.

Page 7, line 6. What measures were taken to ensure meaning was not lost in the notes translation?

Page 7, lines 19-22. it will be key to provide details on how you determined the level of agreement. If you used NVIVO to determine this, provide the output in a supplementary file

The voice of participants is missing in some results sections. For instance, the first paragraph of acceptability and perceived effectiveness lacks supporting quotes to give the participants voice despite presenting essential and interesting findings.

Page 15, line 29-36: Unlike other PM+ studies showing small but positive effects on PTSD, anxiety, or depression (Rahman et al., 2016; Rahman et al., 2019; Jordans et al., 2021), this study found no significant reduction in these symptoms, but a significant decrease in functional impairment. The statement is out of context, discussing RCT Results and not well related with the results of this qualitative paper. Revise.

Discussion section

The discussion section can be written without necessarily repeating the findings.

Line 17, page 19: do you think the results will be generalizable even with a larger sample size? And with qual, how large would this sample size be for it to be generalizable? Generally, with qualitative studies, the aim is never for generalizable results but for applicability in related settings.